# Oral bacteria induce IgA autoantibodies against a mesangial protein in IgA nephropathy model mice

Mizuki Higashiyama[1] , Kei Haniuda[1], Yoshihito Nihei[1,2], Saiko Kazuno[3], Mika Kikkawa[3], Yoshiki Miura[3] , Yusuke Suzuki[2], Daisuke Kitamura[1]

IgA nephropathy (IgAN) is caused by deposition of IgA in the glomerular mesangium. The mechanism of selective deposition and production of IgA is unclear; however, we recently identified the involvement of IgA autoantibodies. Here, we show that CBX3 is another self-antigen for IgA in gddY mice, a spontaneous IgAN model, and in IgAN patients. A recombinant antibody derived from gddY mice bound to CBX3 expressed on the mesangial cell surface in vitro and to glomeruli in vivo. An elemental diet and antibiotic treatment decreased the levels of autoantibodies and IgAN symptoms in gddY mice. Serum IgA and the recombinant antibody from gddY mice also bound to oral bacteria of the mice and binding was competed with CBX3. One species of oral bacteria was markedly decreased in elemental diet-fed gddY mice and induced anti-CBX3 antibody in normal mice upon immunization. These data suggest that particular oral bacteria generate immune responses to produce IgA that cross-reacts with mesangial cells to initiate IgAN.

## Introduction

Autoimmune diseases include a variety of systemic and/or organ-specific diseases, such as systemic lupus erythematosus (Tsokos, 2020) and rheumatoid arthritis (Guo et al, 2018), each exhibiting diverse symptoms caused by a dysregulated immune system that attacks normally tolerated tissues. Pathogenesis also varies among diseases, including tissue inflammation caused by complex reactions of immune cells, such as self-reactive T cells and granulocytes, and/or diverse autoantibodies (auto-Abs). At the other end of the auto-Ab spectrum, organ dysfunction is caused by organ-specific auto-Abs, such as in myasthenia gravis (Conti-Fine et al, 2006) and Graves' disease (Matthews & Syed, 2011). Thus, the mechanisms of development and progression of autoimmune diseases are better understood than ever, but the exact mechanisms triggering disease onset, namely, how self-reactive T cells or B cells producing auto-Abs are generated, remain unclear. Some autoimmune diseases develop after infection with certain viruses or bacteria; thus, it is presumed that T and/or B cells primed by such viruses or bacteria cross-react with molecules expressed in host tissues. In the case of B cells, when the epitopes on the pathogens mimic moieties on a host molecule, such as proteins or glycans, the antibodies (Abs) produced from the primed B cells may cross-react with these host molecules, resulting in autoimmune reactions. In a few autoimmune diseases, "molecular mimicry" is known to cause disease onset, such as Guillain–Barre syndrome and rheumatic fever caused by *Campylobacter jejuni* and *Streptococcus pyogenes*, respectively (Laman et al, 2022). Molecular mimicry of T-cell epitopes between pathogens and host tissues has also been proposed in autoimmune diseases, such as type I diabetes and multiple sclerosis (Wucherpfennig & Strominger, 1995; Kukreja & Maclaren, 2000). However, a causal relationship has only been established in a limited number of cases, and the detailed mechanisms underlying molecular mimicry and autoimmune diseases are not fully understood.

IgA nephropathy (IgAN) is the most common type of primary glomerulonephritis worldwide, with typical symptoms such as proteinuria and hematuria, which gradually progress to end-stage renal failure. Pathologically, IgAN is characterized by the deposition of IgA, IgG, and complement C3 in the glomerular mesangium, followed by mesangial cell proliferation and matrix accumulation in the glomeruli, and finally by destruction of the glomeruli (Wyatt & Julian, 2013). IgAN pathogenesis is attributed to galactose-deficient (Gd)-IgA, which triggers the generation of autoantibodies (auto-Abs) and forms immune complexes (ICs). However, such ICs would not only be deposited in the mesangial region but also at various locations in the glomeruli. In addition, Gd-IgA can be found in the sera of healthy individuals (Chang & Li, 2020). Therefore, Gd-IgA alone cannot explain the mesangium-specific IgA deposition in IgAN. Thus, it is currently unclear why IgA is selectively deposited in the mesangium, and how it is produced.

[1]Division of Cancer Cell Biology, Research Institute for Biomedical Sciences (RIBS), Tokyo University of Science, Noda, Japan [2]Department of Nephrology, Juntendo University Faculty of Medicine, Tokyo, Japan [3]Laboratory of Proteomics and Biomolecular Science, Biomedical Research Core Facilities, Juntendo University Graduate School of Medicine, Tokyo, Japan

Correspondence: kitamura@rs.tus.ac.jp
Kei Haniuda's present address is Department of Immunology, University of Toronto, Toronto, Canada

To help answer these questions, we have been studying an authentic IgAN mouse model, "gddY" mice. These mice were generated by selective intercrossing of mice in early onset groups of outbred ddY mice for more than 20 generations. The original ddY mice are known to develop IgAN, but with low incidence and variable timing of onset. In contrast, essentially all gddY mice exhibit proteinuria and IgA deposition in the glomerular mesangium by 8 wk of age, followed by glomerular injury resembling that observed in human IgAN (Okazaki et al, 2012). Using gene editing in gddY mice, we recently demonstrated that IgA is initially deposited in the mesangium before forming ICs with IgG, IgM, and complement C3 in situ and that the IC formation is necessary for subsequent renal injury, indicating that IgA deposition is an initial event in the pathogenesis of IgAN (Takahata et al, 2020). An IgAN susceptibility gene is located in a chromosomal region that is syntenic with a candidate gene for human familial IgAN (*IGAN1*) (Suzuki et al, 2005). Thus, gddY mice are a useful animal model for studying IgAN pathogenesis. In addition to gddY mice, a few genetically modified mouse models for IgAN have been studied. β-1,4-galactosyltransferase-I (β4-GalT-I)–knockout mice spontaneously develop severe IgAN with glomerular IgA-IC deposition; however, galactose deficiency in many glycoproteins besides IgA complicates the pathogenesis of IgAN in this model (Nishie et al, 2007). The BAFF-transgenic (BAFF-Tg) mouse strain is another IgAN model that exhibits high circulating levels of abnormally glycosylated IgA (McCarthy et al, 2011). However, in these models, the disease is caused by a modified gene, which may not be involved in the pathogenesis of IgAN in humans.

Recently, we found anti-mesangium IgA auto-Abs in the sera of gddY mice and human patients with IgAN and identified βII-spectrin as a target antigen in both gddY mice and IgAN patients. IgA$^+$ plasmablasts (PBs) accumulated in the kidneys of gddY mice and IgA Abs secreted by these PBs, as well as IgA-recombinant monoclonal Abs (rmAbs) generated from them, bound to the mesangium and βII-spectrin (Nihei et al, 2023a). Therefore, we propose that IgAN is a tissue-specific autoimmune disease. In this study, we sought to identify additional self-antigens recognized by other IgA auto-Abs derived from the kidney PBs of gddY mice. We identified the protein CBX3, which stands for Chromobox homolog 3, as a new self-antigen recognized by an rmAb (rmAb#66), which is also bound to oral commensal bacteria of gddY mice, specifically a previously unknown bacterial species that we tentatively called "C42." The binding of rmAb#66 to C42 was competed by CBX3, suggesting molecular mimicry between C42 and CBX3. Our results indicate that particular strains of oral commensal bacteria can induce an immune response that leads to the production of anti-mesangium IgA auto-Abs in gddY mice, which will facilitate the understanding of IgAN pathogenesis and therapeutic strategies for IgAN.

## Results

### Identification of antigens recognized by IgA auto-Abs produced in gddY mice

We recently reported that anti-mesangium IgA auto-Abs are present in the sera of gddY mice and are produced by IgA$^+$ plasmablasts (PBs) that accumulate in the kidneys of these mice. Using

serum IgA, we identified βII-spectrin (also called Sptbn1 for spectrin β chain non-erythrocytic 1) as a self-antigen recognized by these IgA auto-Abs. To identify other self-antigens recognized by IgA auto-Abs, we molecularly cloned IgA Abs produced by PBs in the kidneys of gddY mice, as described previously (Nihei et al, 2023a). cDNAs encoding the $V_H$ and $V_L$ regions of IgA amplified from single IgA$^+$ PBs were recombined with human Cγ1 and Cκ cDNAs, respectively. We expressed each pair of recombinant heavy and light chain genes in HEK293T cells to generate a panel of rmAbs and also generated a 4-hydroxy-3-nitrophenyl acetyl (NP)–specific rmAb with human Cγ1 (#NP) as a negative control. Among the 47 rmAbs tested by Western blot (WB) analysis, six detected specific protein bands, not detected by #NP, in the lysates of primary cultured mesangial cells (MCs) of gddY mice (Fig 1A). Among these rmAbs, we further analyzed rmAb No. 66 (rmAb#66), which recognizes proteins with molecular weights of ~23 and 20 kD.

To identify the antigens recognized by rmAb#66, extracts of MCs of gddY mice were immunoprecipitated with rmAb#66. The precipitates were resolved by SDS–PAGE, and protein bands of similar size to those detected with rmAb#66 by WB were excised and analyzed by mass spectrometry. The peptide sequences repeatedly identified in the immunoprecipitant of rmAb#66, but not rmAb#NP, were used to deduce proteins as candidate autoantigens (Fig 1B). Candidate proteins were evaluated by cDNA cloning and expression. Among them, we identified CBX1, CBX3, and CBX5, also known as HP1β, HP1γ, and HP1α, respectively, as proteins recognized by rmAb#66 (Fig 1C). CBX (chromobox homolog) proteins, alternatively termed heterochromatin protein 1 (HP1), are a family of nuclear heterochromatic adaptor molecules that interact with the methyl groups of histone H3 at lysine 9 (H3K9Me3) and are involved in both gene silencing and heterochromatin formation (Maison & Almouzni, 2004; Sun et al, 2017). These three proteins contain a conserved amino-terminal chromodomain, a variable hinge region, and a conserved carboxy-terminal chromoshadow domain (Bannister et al, 2001; Lachner et al, 2001). Screening of rmAbs by ELISA for binding to CBX proteins revealed that #66 and #71 bound strongly to all three CBX proteins, with the highest reactivity of #66 to CBX3. In addition, #36 bound to CBX3 alone and #25 weakly bound to all three (Fig 1D). rmAb#66 did not bind to dsDNA, histones, LPS, or insulin, which are typical antigens used to assess polyreactive Abs (Fig S1A). Although rmAb#66 $V_H$ and $V_L$ regions contained 4 and 3 amino acid replacements, respectively, due to somatic hypermutation, as observed in most of the other rmAbs from gddY mice, restoration of the mutated sequences to the germline sequence in rmAb#66 did not affect its binding to CBX3 (Fig S1B).

Next, we tested whether serum IgAs from gddY mice reacted with CBX proteins. WB analysis demonstrated that 64% of gddY mice possessed IgA that bound to CBX3, with 57% bound to both CBX3 and CBX1, and 14% bound to all three proteins (Fig 1E). We further examined whether the sera of patients with IgAN also contained IgA that reacted with CBX3 protein. ELISA using human CBX3 protein purified from transfected cells as a coating antigen demonstrated that the sera of 12 of 70 (~17%) patients with IgAN contained anti-CBX3 IgA, based on a cutoff value of the 99% confidence interval (CI) of healthy controls (Fig 1F). A similar result was obtained when the data were limited to age-matched donors (30–49 yr) (Fig S1C; left), with slightly higher anti-CBX3 IgA levels in males than in females

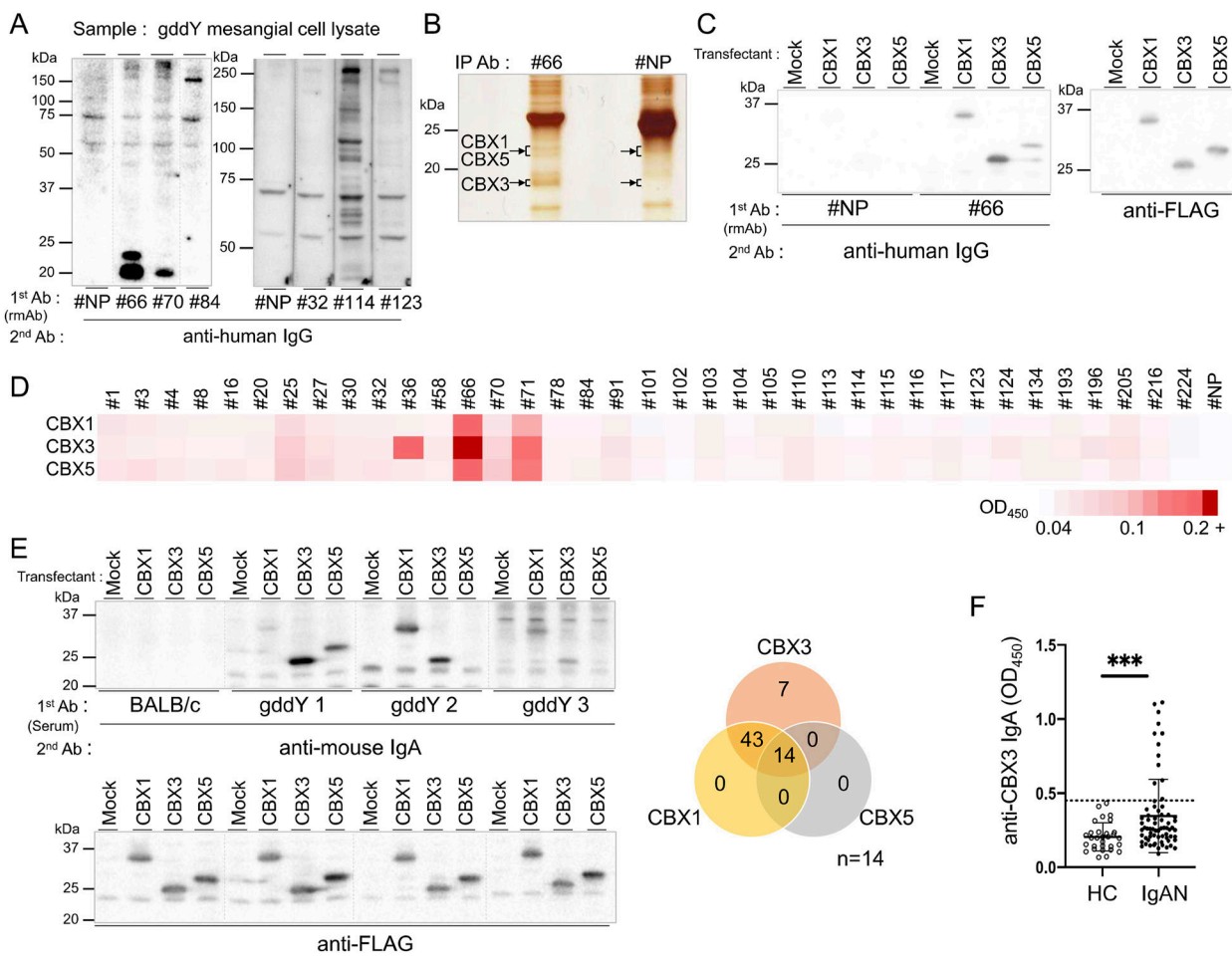

**Figure 1. Identification of MC antigens reactive to IgA auto-Abs generated in gddY mice.**
**(A)** Detection of antigens for rmAbs derived from gddY mouse PBs in the lysates of gddY mouse primary MCs by WB. Split membranes, each containing proteins in one lane, were incubated with the indicated rmAbs (1 µg/ml), followed by an anti-human IgG Ab. **(B)** Immunoprecipitation of gddY mouse MC lysates with rmAb#66 or the control rmAb#NP. SDS–PAGE of the precipitates was developed using silver staining. The bands indicated by arrows were excised and analyzed using mass spectrometry. Those from the rmAb#66 precipitate included CBX1, CBX3, and CBX5, as indicated by arrows. **(C)** FLAG-tagged CBX1, CBX3, or CBX5 proteins transiently expressed in HEK293T cells were detected by WB with the indicated rmAbs (1 µg/ml), followed by anti-human IgG Ab (left panel) or anti-FLAG Ab (right panel). **(D)** ELISA screening of rmAbs for binding to CBX proteins. The heatmap represents the resulting OD values of the indicated rmAbs, with "+" indicating a saturated signal. **(E)** Reactivity of gddY mouse serum IgA with the CBX proteins. WB was performed as in (C) using sera of individual BALB/c or gddY mice as the primary Ab as indicated and anti-IgA Ab as the secondary Ab (left, top) or anti-FLAG Ab to confirm the presence of CBX proteins on the blot (left, bottom). Representative data are shown here. Venn diagram showing the percentage of gddY mice whose serum IgA reacted with CBX1, CBX3, or CBX5 (right panel) (n = 14). **(F)** ELISA determination of anti-CBX3 IgA Abs in sera from healthy controls (HC; n = 30) and patients with IgAN (IgAN; n = 70). ∗∗∗: 0.0006 (Mann–Whitney $U$ test). The dashed line shows 99% confidence interval (CI) of the control sera. One of two or three independent experiments with similar results is shown in each panel.
Source data are available for this figure.

among the patients (Fig S1C; right). Serum IgA binding to CBX3 was also detected by WB analysis in some patient samples (Fig S1D).

## The anti-CBX3 IgA auto-Ab derived from gddY mice binds to the mesangial cell surface

To clarify how auto-Abs recognize CBX proteins, which are generally known to localize to the nucleus, we isolated glomerular cells from the kidneys of BALB/c mice and stained these cells with rmAb#66 or anti-CBX Abs along with other Abs. MCs, endothelial cells, and podocytes were gated based on the expression of CD31 and CD73 (Hatje et al, 2021), and the binding of rmAb#66 or anti-CBX Abs to these cells was analyzed by flow cytometry (FCM). We found that

rmAb#66 bound to ex vivo MCs in the non-permeabilized state, but not to endothelial cells or podocytes, as compared with the negative control rmAb#NP (Fig 2A). In addition, commercially available anti-CBX1 and anti-CBX3 Abs bound only to MCs on their surfaces (Fig 2B). Cell surface anti-CBX3 Ab staining was substantially reduced in MCs from tamoxifen-induced CBX3-conditional knockout mice, in which *CBX3* gene deletion (tail DNA) and CBX3 protein reduction (blood leukocytes) were verified (Fig S2A–C). These results indicate that rmAb#66 recognizes CBX1 and CBX3 expressed on the surface of MCs but not on the surface of endothelial cells or podocytes.

To assess whether the IgA auto-Ab binds to the cell surface of intact glomerular mesangium in vivo, we changed

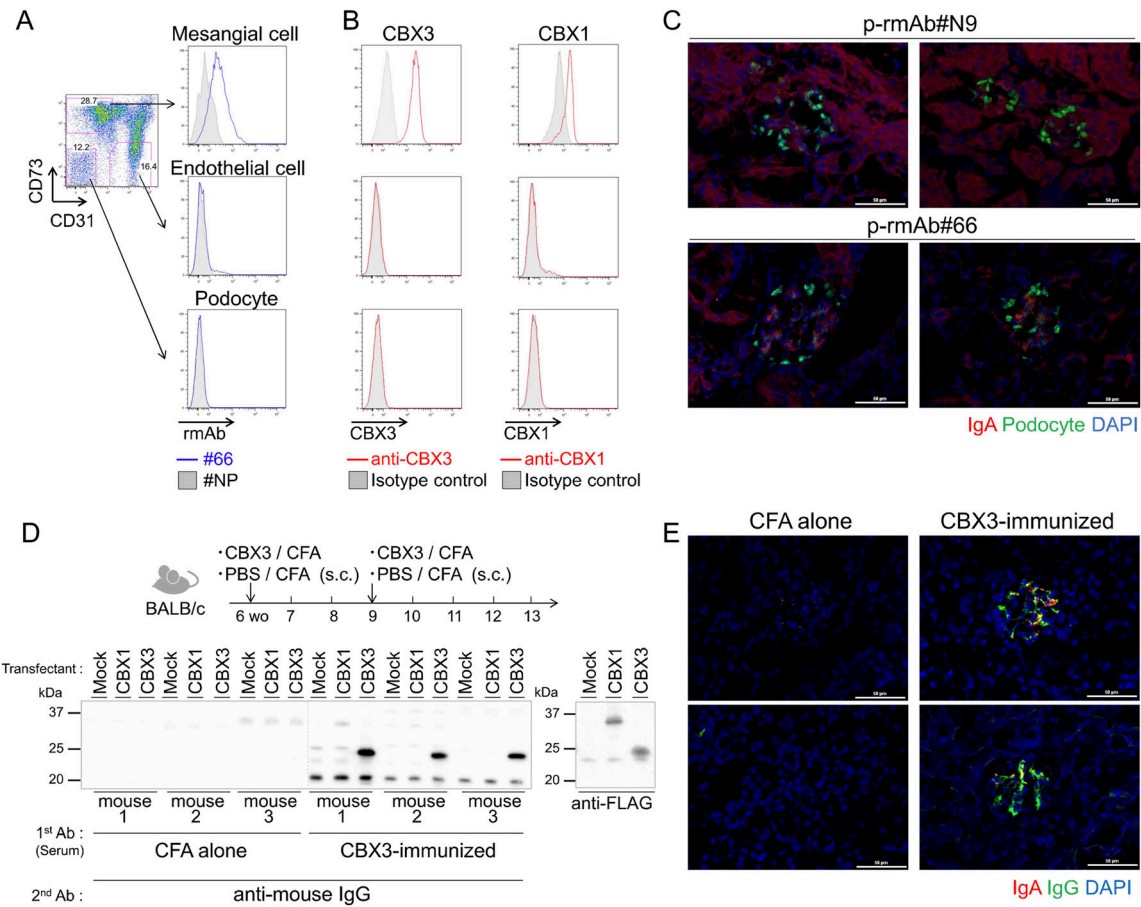

**Figure 2. CBX3 is an autoantigen expressed on the surfaces of MCs.**
**(A, B)** Representative histograms of FCM analysis of glomerular CD45$^-$ cells from BALB/c mice stained with Abs against CD45, CD73, and CD31 for gating on MCs (CD73$^+$ CD31$^-$), endothelial cells (CD73$^-$ CD31$^+$), and podocytes (CD73$^-$ CD31$^-$), and with rmAbs (#66 or #NP) (A) or with anti-CBX3 Ab or anti-CBX1 Abs along with isotype-matched control Ab (rabbit IgG), as indicated (B). **(C)** IFM of kidney sections of AID-knockout mice intravenously injected with dimeric/polymeric control rmAb#N9 (p-rmAb#N9) or CBX-reactive p-rmAb#66 (300 μg per mouse) 2 h before kidney excision, stained with anti-IgA (red), anti-WT1 (indicating podocytes; green), and DAPI (blue). Two representative images (n = 2) for each p-rmAb are shown. White lines indicate scale bars (50 μm). **(D, E)** Anti-glomerular autoantibody production in mice immunized with CBX3. BALB/c mice were immunized s.c. with recombinant CBX3 protein with CFA or CFA alone at 6 and 9 wk of age. **(D)** Lysates of HEK293T cells transfected with FLAG-CBX1 (CBX1), FLAG-CBX3 (CBX3), or empty (mock) vectors were analyzed by WB with sera collected at 8 wk of age as the primary Ab and anti-IgG Ab as the secondary Ab (left panel), or with the positive control anti-FLAG Ab (right panel). **(E)** Kidneys were excised from the immunized mice at 13 wk of age. Kidney sections were stained with anti-IgA (red), anti-IgG (green), and DAPI (blue). Two representative IFM images (n = 2) for different immunizations are shown. White lines indicate scale bars (50 μm). One of two or three independent experiments with similar results is shown in each panel.
Source data are available for this figure.

the C region of the rmAb from human-IgG1 to mouse-IgA and made these rmAbs dimeric/polymeric by expressing them in HEK293T cells stably transfected with the J chain (Nihei et al, 2023a). The resultant dimeric/polymeric rmAb#66 (p-rmAb#66) or control rmAb (p-rmAb#N9), the latter containing V regions of IgM from a randomly picked single B cell from gddY mouse spleen, was intravenously injected into AID-deficient mice lacking endogenous class-switched Abs. 2 h after injection, the kidney was excised, and p-rmAb deposition was analyzed by immunofluorescence (IF) staining of the sections. The results demonstrated that p-rmAb#66, but not p-rmAb#N9, specifically bound to the glomeruli in a non-podocyte area (Figs 2C and S2D), suggesting that p-rmAb#66 reacts in situ with CBX1 or CBX3 exposed on the surface of cells in glomeruli.

To examine whether CBX3 indeed behaves as an autoantigen in the glomerular mesangium, we immunized normal BALB/c mice with CBX3 protein in CFA. WB analysis demonstrated that the serum from CBX3-immunized mice contained anti-CBX3 IgG Abs (Fig 2D). 4 wk after secondary immunization, the kidney was excised, and Ab deposition was analyzed by IF staining of the sections. The results demonstrated that IgG and IgA were specifically deposited in the mesangial regions of glomeruli in CBX3-immunized mice but not in mice administered CFA alone (Fig 2E). No significant deposition of IgG or IgA was detected in organs other than the kidney of mice immunized with CBX3 (Fig S2E), although IgA$^+$ plasma cells, known to reside in the liver, were stained with anti-IgA in both groups as previously reported (Moro-Sibilot et al, 2016). These results indicated that in vivo–induced Abs against CBX3 specifically bind to the glomerular mesangium.

### Diet is involved in auto-Ab production and development of IgAN in gddY mice

Some studies have shown that some patients with IgAN have IgA Abs against gliadin, a glycoprotein component of gluten contained in wheat, which is known to generate auto-Abs that attack the small intestine when ingested by patients with celiac disease (Fornasieri et al, 1987; Collin et al, 2002). Although the relationship between celiac disease and IgAN remains unclear, we focused on the involvement of dietary antigens in auto-Ab induction in IgAN. We fed gddY mice for 7 wk an elemental diet (ED) composed of amino acids and dextrin but lacking proteins, excised the kidneys, and analyzed Ab deposition by IF staining of the sections. The results demonstrated that deposition of IgA and IgG was hardly observed in gddY mice fed an ED (Fig 3A). Moreover, the proliferation of MCs (Fig 3B) and renal damage indicated by albuminuria (Fig 3C) and blood urea nitrogen (BUN) (Fig 3D) were reduced in the ED-fed mice. In addition, WB analysis showed the disappearance of IgA against CBX1 and CBX3 in the serum of ED-fed gddY mice (Fig 3E). These data strongly suggest that some dietary components missing in the ED and/or digestive system environment established by a normal diet, but not by ED, are involved in the induction of anti-CBX IgA auto-Abs and pathogenicity in gddY mice. Dietary protein itself was not likely responsible for these differences because gddY mice fed an amino acid diet (AAD) that also lacked protein did not significantly affect albuminuria (Fig S3A).

### Antibiotic treatment suppresses auto-Ab production and development of IgAN in gddY mice

Dietary components substantially affect the composition of commensal bacteria (De Filippo et al, 2010; Sonnenburg et al, 2016). We hypothesized that a change in bacterial composition due to ED feeding might suppress auto-Ab production and improve IgAN in gddY mice. To test this, we administered gddY mice an antibiotic cocktail (ABX) in the drinking water from 4 to 6 wk of age and analyzed these mice at 7 wk of age. The deposition of IgA and IgG was diminished in mice administered ABX, as demonstrated by IF staining of kidney sections (Figs 4A and S3B). In addition, albuminuria (Fig 4B) was attenuated by ABX administration. Furthermore, WB analysis revealed the disappearance of IgA auto-Abs against CBX1 and CBX3 in the sera of the treated gddY mice (Fig 4C). These data indicated the involvement of commensal bacteria in the development of IgAN and auto-Ab production in gddY mice.

### Auto-Abs of gddY mice react with commensal bacteria in their oral cavity

Auto-Abs that cross-react with components of commensal bacteria have recently been reported (Ruff et al, 2019; Girdhar et al, 2022). Given the disappearance of IgA auto-Abs in ABX-treated gddY mice, we speculated that B cells initially primed with commensal bacteria differentiate into IgA-producing PBs and that such IgA cross-reacts with self-antigens on MCs, resulting in mesangial IgA deposition. Accordingly, WB analysis demonstrated that serum IgA of gddY mice bound to proteins in cultured bacteria from the oral cavity but not to those from the feces or small intestine (Figs 5A and S4A).

Furthermore, IgA in the culture supernatant of leukocytes (including PBs) from gddY kidneys selectively bound to the proteins in the lysates of oral bacteria (Figs 5B and S4B). FCM analysis showed that serum IgA from gddY mice bound to a portion of the oral bacteria but not to those from the small intestine of gddY mice (Fig 5C; gating strategy is shown in Fig S4C) and not to the oral bacteria of BALB/c mice (Fig 5D). Serum IgA from BALB/c mice did not bind to the oral bacteria of gddY mice, indicating that the binding of gddY mouse serum IgA to the same bacteria is specific (Fig S4D). IgA in the culture supernatant of leukocytes from the kidney, but not from the small intestine of gddY mice, bound to oral bacteria from gddY mice, as shown by FCM analysis (Fig 5E). Interestingly, IgA from salivary glands bound to a much higher proportion of oral bacteria than IgA from the kidney, suggesting that the antibacterial repertoire of IgA Abs from PBs localized in the salivary glands is much broader than that from PBs in the kidneys of gddY mice.

Similar to the serum Ab, rmAb#66 reacted with a portion of oral bacteria, especially anaerobes, but with very few fecal bacteria in gddY mice (Fig 5F). These results indicate that rmAb#66 is reactive not only to CBX proteins but also to surface molecule(s) of oral bacteria, indicating that gddY mice produce IgA that cross-reacts with both mesangial proteins and oral bacteria.

To verify the cross-reactivity of rmAb#66 with CBX proteins and oral bacteria, we tested whether its binding to oral bacteria could be inhibited by the addition of the CBX3 protein. As expected, preincubation with CBX3, but not OVA, inhibited the binding of rmAb#66 to oral bacteria (Fig 5G). This result indicates that rmAb#66 recognizes an epitope shared by CBX3 and some molecule(s) on the surface of oral bacteria.

### Identification of a bacterial strain that shares an rmAb#66 epitope with CBX3

Next, we sought to identify the bacteria that were recognized by rmAb#66. We randomly selected 61 colonies of anaerobic bacteria from the oral cavity of gddY mice, stained them with rmAb#66, and analyzed them by FCM. Among these, 12 were positive for rmAb#66 staining. They shared the identical 16S rRNA sequence that had the highest similarity (98.66%) to, but distinct from, that of *Streptococcus danieliae*, and here we tentatively name it by its colony number "C42." FCM analysis confirmed that rmAb#66 bound to C42, albeit only partially (Fig 6A). As in the case of bulk oral bacteria, the binding of rmAb#66 to C42 could be inhibited competitively by the CBX3 protein, but not by ovalbumin (Fig 6B) or chicken γ globulin (CGG) (Fig S5). Since PNGase F treatment of C42 did not affect the binding of rmAb#66, the epitope on C42 was not likely to be a glycan (Fig 6C). To clarify the possibility that C42 shares antigenicity with CBX3, we immunized BALB/c mice with UV-killed C42 or an unrelated strain (*E. faecalis*) that did not bind to rmAb#66. 4 wk after immunization, sera from mice immunized with C42 but not with *E. faecalis* contained IgG bound to the CBX3 protein (Fig 6D). In addition, IgA deposition on glomeruli was observed in the kidneys of the former mice, but not in the kidneys of the latter mice (Fig 6E). Collectively, these data suggest that the production of anti-mesangium IgA auto-Abs is induced by an immune response to cross-reactive bacteria residing in the oral cavity of gddY mice.

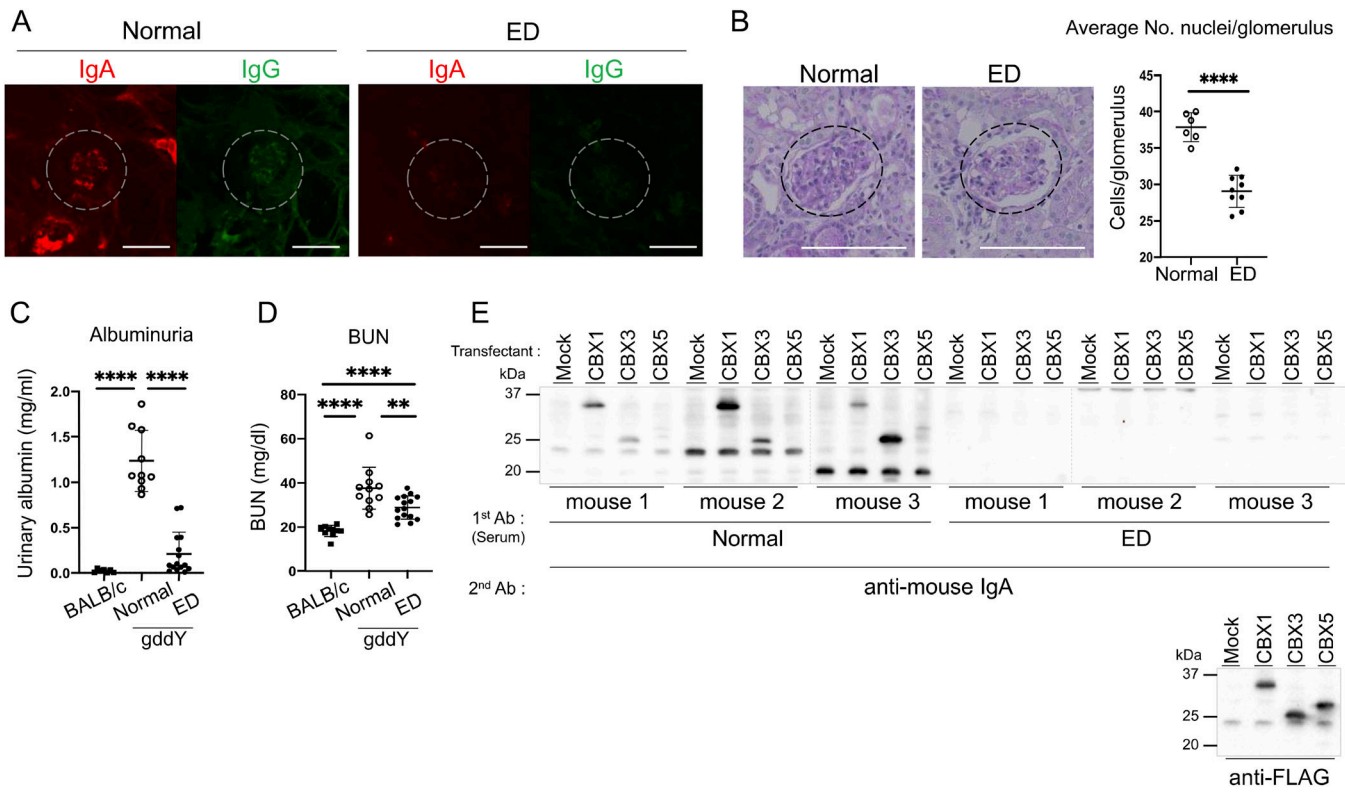

**Figure 3. Amelioration of IgAN with reduced autoantibodies in gddY mice using an elemental diet.**
In all experiments, gddY mice were fed an ED or a normal feed from 2 wk of age. **(A)** Kidney sections excised at 9 wk of age were stained with anti-IgA (red) and anti-IgG (green) Abs. Representative IFM images are shown. Dashed circles indicate areas of glomeruli and white lines indicate scale bars (100 μm). **(B)** PAS staining of the kidney sections used in (A). Dashed circles indicate areas of glomeruli and white lines indicate scale bars (100 μm) (left panels). The average number of nuclei per glomerulus, counted visually in 20 glomeruli per mouse, is plotted in the right panel (normal diet, n = 6; ED, n = 9). **(C, D)** Urinary albumin (C) and BUN (D) levels in gddY mice fed the same diet and normally fed BALB/c mice at 8 wk of age (BALB/c mice, n = 10; normal diet, n = 11; ED, n = 15). **(E)** WB analysis was performed as shown in Fig 1E with sera from treated gddY mice used as primary Abs. **(B, C, D)** Data are presented as mean ± SD of biological replicates. **P < 0.01; ****P < 0.001. P-values were calculated using the unpaired t test. One of two or three independent experiments with similar results is shown in each panel.
Source data are available for this figure.

In conjunction with the results of ABX administration (Fig 4), the above results suggest that the reduction of IgAN symptoms and disappearance of serum anti-CBX3 IgA in gddY mice fed an ED (Fig 3) may be due to alterations in oral commensal bacteria. To support this idea, FCM analysis revealed that rmAb#66-binding bacteria disappeared from the oral cavity of ED-fed gddY mice (Fig 6F). The frequency of C42 (OTU002), as determined by sequencing of the V3 and V4 regions of the 16s rRNA gene, among bacteria in the oral cavity was significantly reduced in gddY mice fed an ED diet. Another Streptococcus strain, OTU004, also decreased after ED feeding, although it was less dominant than C42 among the oral bacteria (Fig 6G). Taken together, these data indicate that commensal bacteria represented by C42, selectively increased in the oral cavity of gddY mice, may induce an immune response that produces IgA that is cross-reactive with CBX3 and binds to MCs of glomeruli.

### Determination of the rmAb#66 epitope on CBX3

To narrow down the region of CBX3 that is recognized by rmAb#66, we made overlapping fragments of CBX3 protein, namely 1/2 CBX3 (AA1-92) and 2/3 CBX3 (AA63-184), and tested rmAb#66 binding by WB (Fig 7A).

The results clearly showed that rmAb#66 bound to 1/2 CBX3, but not to 2/3 CBX3 (Fig 7B). In addition, the serum IgA of gddY mice bound to 1/2 CBX3 (Fig 7C). To identify the epitope of rmAb#66, we introduced point mutations replacing individual amino acid (AA) with alanine into the 1/2 CBX3 fragment at AA11 methionine (11MA), AA32 valine (32VA), AA54 phenylalanine (54FA), and AA58 aspartic acid (58DA) (Fig 7D). These positions were selected from the regions that did not overlap with 2/3 CBX3 and from stretches of sequences conserved among CBX3, CBX1, and CBX5, all of which bind to rmAb#66. WB analysis revealed that rmAb#66 binds to the mutated 11MA and 58DA, similar to the original 1/2 CBX3 (WT), but weakly to 32VA and not to 54FA (Fig 7D). The loss of reactivity of rmAb#66 to 54FA was verified by ELISA (Fig 7E). These findings strongly suggest that rmAb#66 recognizes an epitope on CBX3 that includes AA54 phenylalanine.

## Discussion

A pathological feature of IgAN is deposition of IgA in the glomerular mesangium. Although half a century has passed since IgAN was identified, it is unclear why IgA is deposited selectively in the

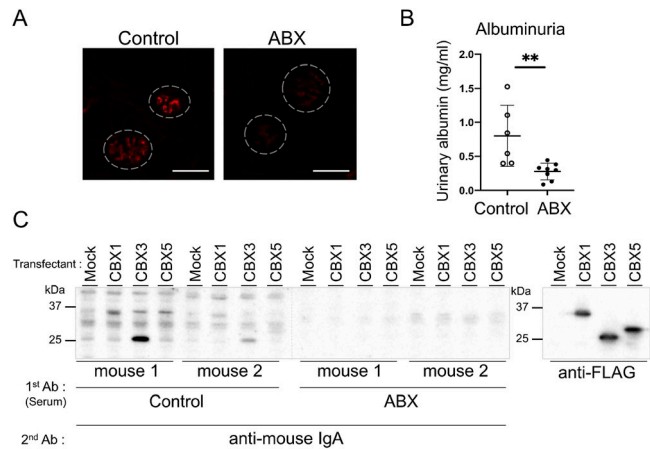

**Figure 4.  Amelioration of IgAN with a reduction in auto-Abs in gddY mice by antibiotic treatment.**

gddY mice (4 wk old) were administered an antibiotic cocktail (ABX; see the Materials and Methods section for composition) in drinking water or water alone (control) for 2 wk. 1 wk after the last day of administration, the kidneys were excised, and sera were sampled for the following analyses. **(A)** Representative IFM images of kidney sections stained with anti-IgA Ab (red). Dashed circles indicate areas of glomeruli and white lines indicate scale bars (100 μm). **(B)** Urinary albumin concentration in each mouse is plotted (control, n = 6; ABX, n = 8). **(C)** WB analysis as in Fig 3E with sera from the gddY mice treated as above used as primary Abs (left panel) or with the positive control anti-FLAG Ab (right panel). **(B)** Data are presented as mean ± SD of biological replicates. **\*\*P < 0.01. P*-values were calculated using the unpaired *t* test. One of two independent experiments with similar results is shown in each panel.
Source data are available for this figure.

mesangium and how such IgA is produced. The data described here suggest that oral bacteria prime immune responses leading to the production of IgA auto-Abs that recognize a protein expressed on the surface of mesangial cells. This finding provides new insights into the pathogenesis and treatment of IgAN.

We identified CBX3 expressed on mesangial cell surfaces as a new self-antigen recognized by an rmAb (rmAb#66) derived from PBs in the kidneys of gddY mice. IgA Abs that bind to the CBX3 protein could also be detected in sera from two-thirds of gddY mice and in ~20% of patients with IgAN. Thus, in addition to βII-spectrin, CBX3 is a common self-antigen for IgA in IgAN. Although CBX proteins are known to be localized in the nucleus, in vitro staining of MCs with rmAb#66 and a commercially available CBX3-specific Ab, as well as the in vivo binding of p-rmAb#66 to the mesangium, revealed cell surface expression of CBX3. Thus, CBX3 on the surface of MCs can be a direct target of IgA auto-Abs in IgAN. Although the mechanism by which CBX3 protein is expressed on MCs remains obscure, we observed that βII-spectrin, which typically forms a cytoskeletal meshwork at the inner plasma membrane in complex with αII-spectrin, tropomyosin, actin, and others, was also expressed on the surface of MCs (Nihei et al, 2023a). Thus, MCs may have the unique ability to translocate particular intracellular proteins to the cell surface. Concerning this possibility, it has been reported that actin is expressed on the surface of nerve cells (Miles et al, 2006). Interestingly, CBX3 was also reported to be present in the cytoplasm interacting with actin in myoblasts (Charó et al, 2018),

which are derived from mesenchymal stem cells, similar to MCs (Avraham et al, 2021). Thus, CBX3 may be expressed on the surface of MCs in conjunction with actin.

Our data showed that a substantial proportion of patients with IgAN and gddY mice had IgA against CBX3 in their sera, suggesting that anti-CBX3 IgA auto-Abs cause IgAN, at least in some cases, as we recently proposed for anti-βII-spectrin IgA (Nihei et al, 2023a). Although CBX3 immunization of BALB/c mice induced anti-CBX3 IgG in the sera and deposition of IgG and IgA in the kidneys, it did not worsen albuminuria in our study (data not shown). This suggests that factors besides the glomerular Ab deposition are necessary for disease onset. First, the genetic background of gddY mice may contribute to pathology. In this regard, we observed that splenic B cells from gddY mice proliferated to a greater extent in response to anti-IgM or anti-CD40 Ab stimulation than those from BALB/c mice (data not shown). Second, anti-CBX3 Abs in the immunized BALB/c mice most likely have normal glycosylation, unlike Gd-IgA commonly detectable in the sera and glomerular deposits in patients with IgAN, which forms ICs with IgG auto-Ab against Gd-IgA (Suzuki et al, 2011a). Gd-IgA may also be critical for IgAN pathogenesis in mice. Indeed, a correlation between aberrantly glycosylated IgA and IgAN onset has been experimentally demonstrated in ddY mice (Makita et al, 2020). Third, it is presumed that cellular components such as T cells, dendritic cells, macrophages, and cytokines and other ligands produced by these cells, also contribute to the induction of an inflammatory response in the mesangium in IgAN (Chang & Li, 2020), which may be substantially different from those activated in the immunized BALB/c mice in our experiment.

We found that commensal bacteria also contributed to the pathogenesis of IgAN in gddY mice, as shown by ABX treatment in mice. In addition, serum IgA from gddY mice bound to commensal bacteria in the oral cavity but not in the intestines of gddY mice. Moreover, IgA secreted from PBs accumulated in the kidneys and salivary glands bound to oral bacteria. These data led us to hypothesize that some bacterial strains in the oral cavity serve as antigens that induce an immune response to generate PBs that produce pathogenic IgA. This hypothesis is further supported by our data showing that a monoclonal rmAb#66 derived from a PB in the gddY mouse kidney cross-reacts with CBX proteins on mesangial cells and with oral bacteria in gddY mice, and a clone "C42" isolated from these bacteria, in a competitive manner. More direct evidence was shown by the result that immunization of BALB/c mice with C42 induced serum Abs that bound to CBX3 in vitro and to glomeruli in vivo. Although the mechanisms by which commensal bacteria spontaneously induce an immune response to produce IgA in gddY mice are still unknown, it is possible that some antigenic epitope on the bacteria happens to mimic an epitope on CBX3; therefore, the IgA produced cross-reacts with this protein on MCs. We are trying to prove such molecular mimicry between the bacterial antigen and self-antigen using C42 and CBX3. We have already identified an rmAb#66 epitope in the latter, as described above (Fig 7). Since C42 is a previously undescribed bacterial strain, whole-genome sequencing will be necessary to generate a whole-protein reference database to identify an antigen and epitope recognized by rmAb#66. Although C42 was derived from a single colony of the bacteria, rmAb#66 binds to only a portion of this bacteria, ranging

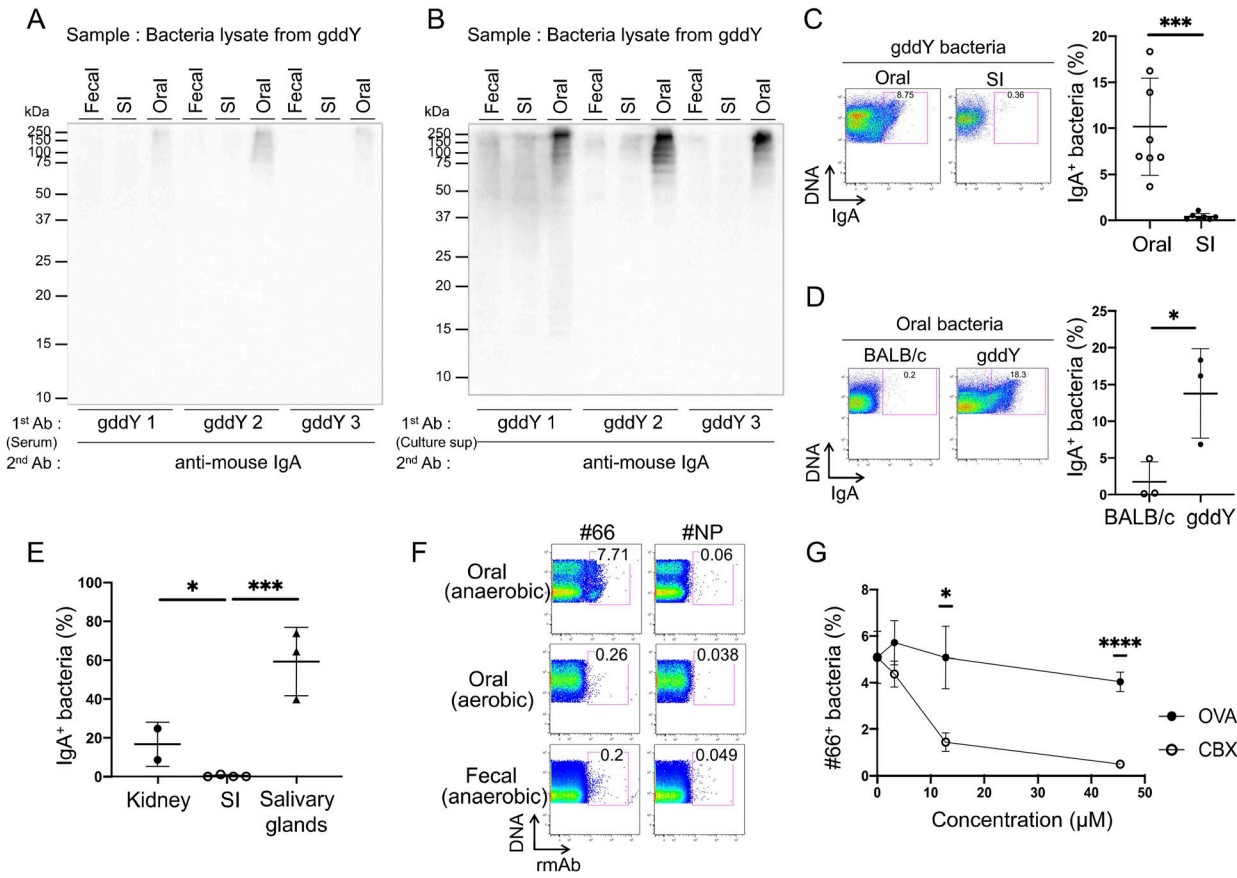

**Figure 5. The gddY auto-Abs reacted with commensal bacteria in the oral cavity of the gddY mice.**
**(A, B)** WB analysis of lysates of bacteria from feces (fecal), small intestine (SI), and oral cavity (oral) of gddY mice blotted with sera from gddY mice (A) or culture supernatant of leukocytes from gddY mouse kidneys (B) as the primary Abs and anti-IgA as the secondary Ab. The IgA concentration in the primary Abs was adjusted to 1 µg/ml by dilution. The bacteria were cultured anaerobically on brain heart infusion agar before lysis. **(C, D, E, F)** FCM analyses of cultured bacteria stained with various Ab solutions (containing 1 µg/ml IgA) and SYBR Green (for bacterial DNA). **(C, D)** Bacteria from the oral cavity (oral) or small intestine (C), or oral bacteria from BALB/c or gddY mice (D), were stained with sera from gddY mice. **(E)** Oral bacteria from gddY mice were stained with the culture supernatant of leukocytes from the indicated organs of gddY mice. **(C, D, E)** APC-conjugated anti-mouse IgA was used as secondary Ab. The secondary Ab alone did not stain cultured bacteria (data not shown). **(F)** Oral or fecal bacteria from gddY mice cultured under the indicated conditions were stained with rmAb#66 or rmAb#NP, followed by Alexa Fluor 647–anti-human IgG. **(C, D, E)** The frequencies of IgA-bound bacteria among whole bacteria stained with SYBR Green are plotted. Data are presented as means ± SD of biological replicates. **(G)** rmAb#66 (1 µg/ml) was incubated with the indicated concentrations of recombinant CBX3 or OVA in equal volumes for 20 min at RT, mixed with cultured oral bacteria from gddY mice, and incubated overnight on ice. The bacteria were stained with Alexa Fluor 647-anti-human IgG to detect bound rmAb#66, together with SYBR Green to label the bacteria, and analyzed by FCM. The frequencies of rmAb#66-bound bacteria among all bacteria were plotted. Data are presented as means ± SD of three technical replicates. *$P < 0.05$; ***$P < 0.005$; ****$P < 0.001$. $P$-values were calculated using unpaired $t$ test. One of two independent experiments with similar results is shown in each panel.
Source data are available for this figure.

from a few percent to 10% depending on the FCM analysis. This suggests that the cell surface expression of the antigen recognized by rmAb#66 may vary among individuals in a colony in a reversible manner, which may be explained by a well-known phenomenon called phase variation (Van Der Woude & Bäumler, 2004).

We observed that feeding gddY mice an ED strikingly suppressed IgAN development and almost completely abolished the production of IgA auto-Abs against the mesangium and CBX proteins. This was not due to the absence of protein content in the ED, as feeding an AAD diet similarly lacking proteins did not affect the extent of albuminuria. We observed a marked reduction in the C42 strain, as determined by the 16s rRNA gene sequence in the oral cavity of gddY mice after ED feeding. Considering the data indicating that C42 may be responsible for priming the immune response to produce anti-mesangial auto-Abs, the decrease in C42 in the oral cavity is likely a reason for the outcome of ED feeding in gddY mice. The decrease in C42 may be due to the difference in ingredients in the ED as compared to normal diets; for example, the former lacks dietary fibers, a main energy source for gut and oral bacteria (König, 2000; Kajiura et al, 2009; Andoh et al, 2019). Alternatively, the liquid form of ED, in contrast to the solid form of a normal diet, may affect the colonization of C42 in the oral cavity because the form of the diet is known to affect the amount of saliva (Ito et al, 2001) and influence bacterial colonization (Carpenter, 2020).

Since IgAN is often exacerbated by infection in the upper respiratory tract and tonsillectomy can improve the disease condition, it has been assumed that oral mucosal immunity is involved in the onset of IgAN (Hotta et al, 2001; Miura et al, 2009; Suzuki et al,

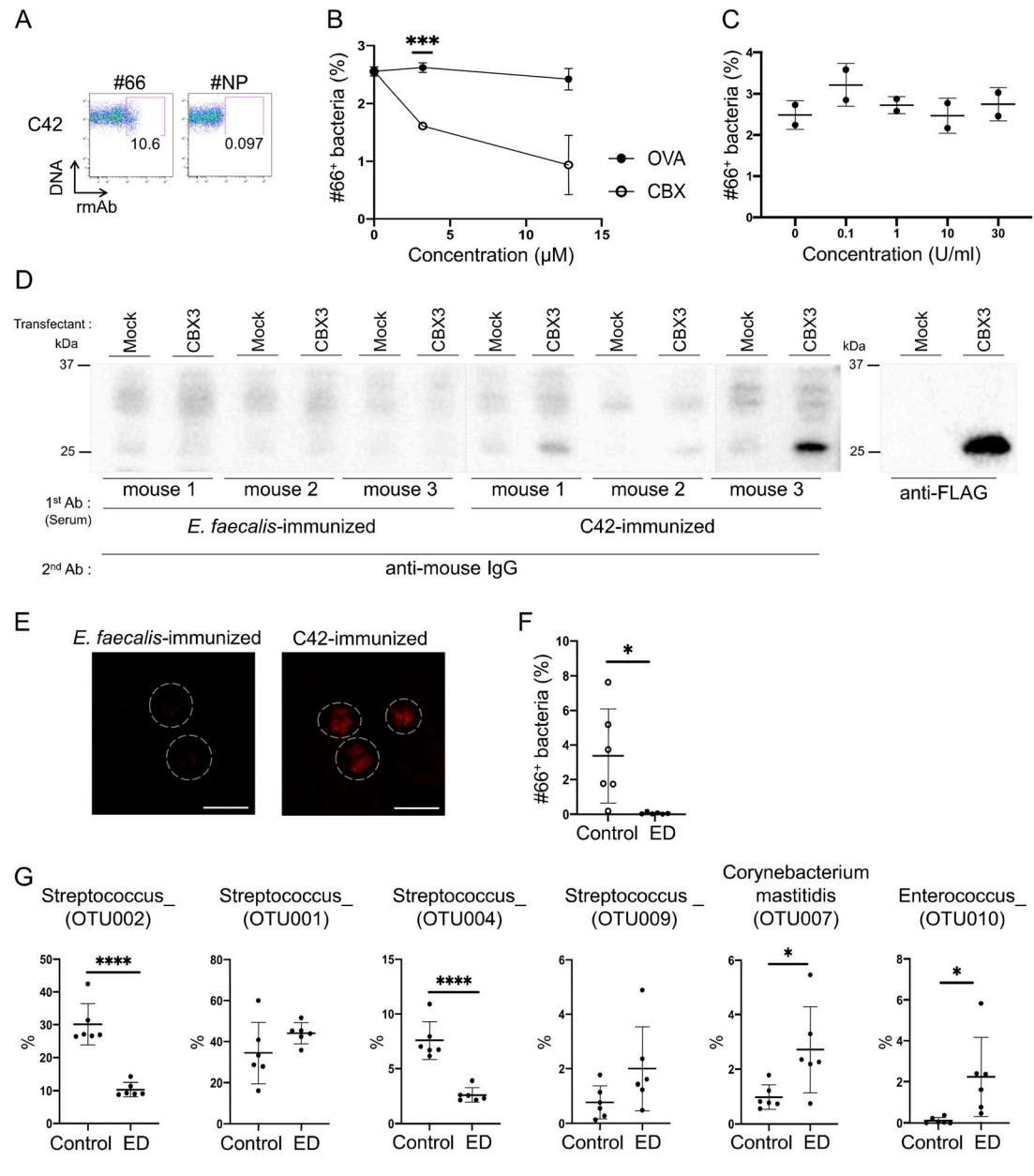

**Figure 6. Identification of a bacterial strain that shares an rmAb#66 epitope with CBX3.**
**(A)** Isolated bacteria (C42) were stained with rmAb#66 or rmAb#NP (1 µg/ml), followed by Alexa Fluor 647–anti-human IgG together with SYBR Green, and analyzed by FCM. **(B)** rmAb#66 (1 µg/ml) was incubated with CBX3 or OVA and then with C42, which was stained with anti-human IgG and analyzed as described in Fig 5G. Data are presented as means ± SD of two technical replicates. **(C)** C42 bacteria (2 × 10$^6$ cells) were incubated with the indicated concentrations of PNGase F (Roche) for 4 h at 37°C, stained with rmAb#66 (1 µg/ml) overnight on ice, and labeled with SYBR Green. Binding of rmAb#66 to C42 was detected by FCM using Alexa Fluor 647-anti-human IgG (n = 2). **(D)** BALB/c mice were immunized s.c. with UV-killed C42 or *E. faecalis*, together with zymosan as an adjuvant, at 4 wk of age. WB analysis was performed as shown in Fig 3E (except that CBX1 and CBX5 samples were omitted) with sera from immunized mice at 8 wk of age as the primary Abs and anti-IgG Ab as the secondary Ab (left panel) or anti-FLAG Ab (right panel). **(E)** Representative IFM images of kidney sections from the mice used in (D). Kidney sections were stained with an anti-IgA Ab (red). Dashed circles indicate areas of glomeruli and white lines indicate scale bars (100 µm). **(F)** Frequency of bacteria recognized by rmAb#66 among all bacteria (SYBR Green$^+$) in the oral cavity of gddY mice (n = 6). Bacteria from gddY mice fed a normal diet (control) or an ED (ED) were cultured anaerobically on brain heart infusion agar, stained with rmAb#66 (1 µg/ml) and Alexa Fluor 647–anti-human IgG, together with SYBR Green, and analyzed by FCM. **(G)** Frequency of each bacterial strain among the total bacteria in the oral cavities of gddY mice fed a normal diet (control) or an ED, as determined by 16S rRNA bulk sequencing (n = 6). An underscore after each genus name indicates that the species was not classified. Data are presented as means ± SD of biological replicates. *P < 0.05; ***P < 0.005; ****P < 0.001. P-values were calculated using the unpaired t test. One of two independent experiments with similar results is shown in each panel.
Source data are available for this figure.

2011b). However, experimental evidence for the role of mucosal immunity in IgAN is still unclear, hampering the global use of tonsillectomy as a treatment for IgAN. Based on our results, we assume that some commensal bacteria in the oral cavity trigger the B-cell response in the mucosal immune system, such as the tonsil, leading to the production of anti-mesangial IgA auto-Abs in

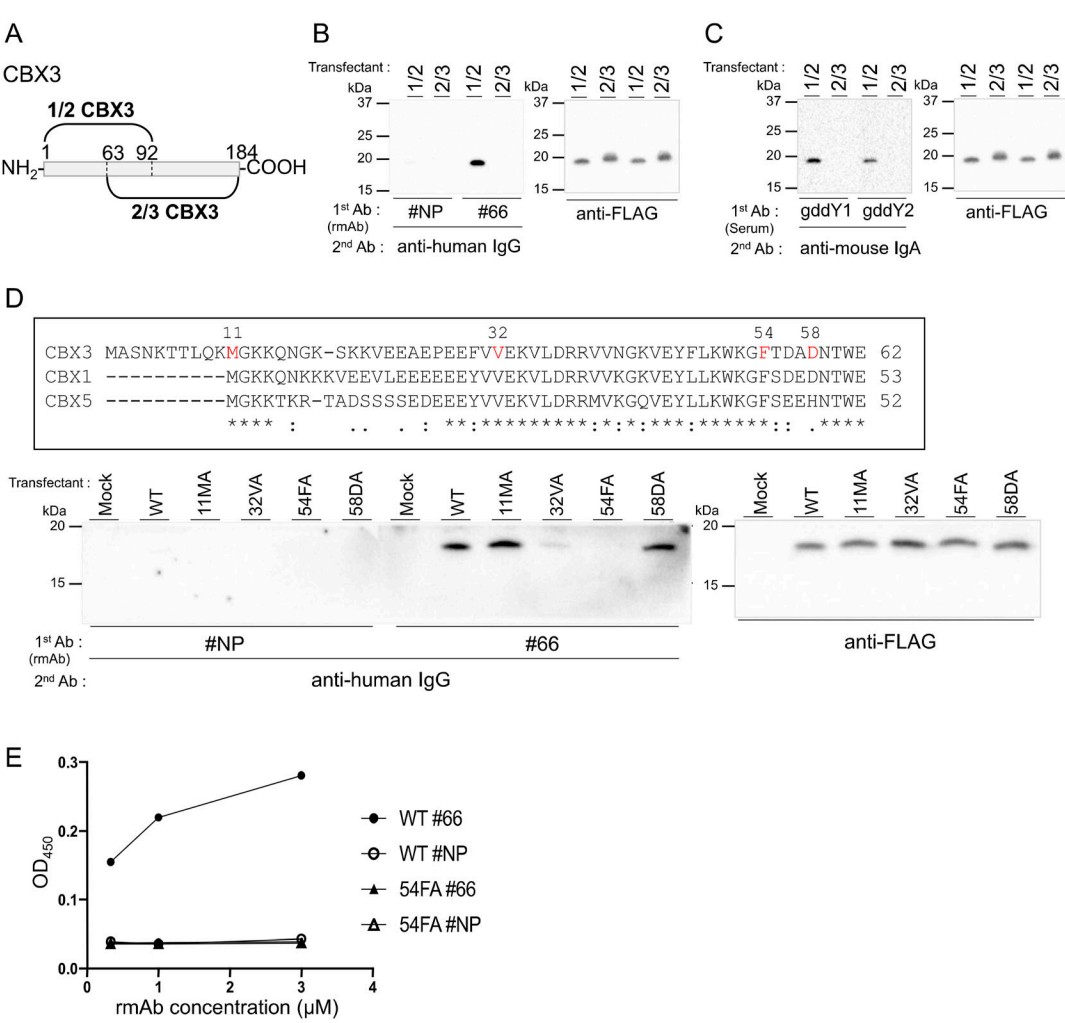

**Figure 7. Determination of rmAb#66 epitope on CBX3.**
**(A)** Schematic representation of CBX3 proteins: 1/2 CBX3 (AA1-92) and 2/3 CBX3 (AA63-184). **(B, C)** FLAG-tagged 1/2 or 2/3 CBX3 proteins, transiently expressed in HEK293T cells, were detected by WB with rmAb#66 or rmAb#NP (1 μg/ml) followed by anti-human IgG Ab ((B), left panel). The same membrane was reblotted with sera of two gddY mice followed by anti-mouse IgA Ab ((C), left panel). Protein loading was confirmed using an anti-FLAG Ab ((B, C), right panels; the same blot is shown). **(D)** Amino acid sequence alignment of CBX proteins (AA1-62) (top). WB analysis of FLAG-tagged WT or the indicated mutated CBX3 proteins transiently expressed in HEK293T cells, performed as in (B) (bottom). **(E)** Reactivity to CBX3 (WT) or CBX3 mutant 54FA (54FA) proteins of the indicated rmAbs at the indicated concentrations was evaluated by ELISA and presented as $OD_{450}$ values. One of two independent experiments with similar results is shown in each panel.
Source data are available for this figure.

humans, resulting in the onset of IgAN. If this is the case, identification of the epitopes shared by the oral bacteria and mesangial self-antigen, both recognized by the same IgA auto-Abs, may lead to the development of reagents mimicking these epitopes in structure, which could bind to the auto-Abs and block their binding to the mesangium, providing a new strategy for the treatment of IgAN.

# Materials and Methods

### Mice

gddY mice were generated as previously described (Okazaki et al, 2012). BALB/c mice were purchased from Sankyo Labo Service

Corporation, Inc. AID-deficient C57BL/6 mice were provided by Dr. Honjo (Wei et al, 2011). $CBX3^{fl/fl}$ (B6; B6129-$Cbx3^{tm1.1Hko}$; RIKEN BRC RBRC09866) $CreER^{T2}$ (10471; Taconic) C57BL/6 mice were provided by Dr. Koseki. Unless otherwise noted, the gddY mice used in this study were 8 wk old. All mice were maintained at the Tokyo University of Science (TUS) mouse facility under specific pathogen-free conditions. Mouse procedures were performed according to the protocols approved by the TUS Animal Care and Use Committee. Normally, mice were fed a solid feed (FR-1; Funabashi Pharm). In the dietary experiments, 2-wk-old gddY mice were fed 1.74 kcal/ml of elemental diet (ED; Elental; EA Pharma) mixed with drinking water or an AAD (A07060801 Research diets) for the indicated time periods. To deplete commensal bacteria, 4-wk-old gddY mice were administered a mixture of antibiotics (ABX; 1 g/liter ampicillin, 0.5 g/liter vancomycin, 1 g/liter neomycin, and 1 g/liter metronidazole) in

drinking water for 2 wk. The mice were analyzed 1 wk after the last administration. Tamoxifen (0.09 g/kg body weight; Sigma-Aldrich) was intraperitoneally administrated to *CBX3*$^{fl/fl}$ *CreER*$^{T2}$ mice once daily for three consecutive days and after 2 d suspension, additional doses were given for two consecutive days. FACS analysis was performed 5 d after the last tamoxifen injection.

## Human subjects

Sera from healthy individuals and patients with IgAN were obtained at Juntendo University Hospital with informed consent from patients and approval from the Research Ethics Review Committees of Juntendo University Hospital. Laboratory information of patients and healthy volunteers was recently reported by our group (Nihei et al, 2023a).

## Antibodies

The following Abs were used for IF microscopy (IFM): anti-mouse IgA Ab-PE (97013; Abcam), anti-mouse IgG Ab-FITC (1030-02; Southern Biotech), rabbit monoclonal anti-Wilms tumor protein Ab (ab89901; Abcam), and anti-rabbit IgG-Alexa Fluor 488 (ab150073; Abcam). For FCM, the following Abs were used: anti-mouse IgA-APC (clone 11-44-2; Southern Biotech), anti-mouse CD73-biotin (clone TY/11.8; BioLegend), anti-mouse CD45-PerCP-Cy5.5 (clone 30-F11; BioLegend), anti-mouse CD31-PE (clone 390; BioLegend), anti-mouse TER119-BV510 (clone TER-119; BioLegend), rabbit polyclonal anti-HP1γ (CBX3) Ab (2619; Cell Signaling Technology), rabbit monoclonal anti-HP1β (CBX1) Ab (8676; Cell Signaling Technology), rabbit IgG (for isotype-matched control; 011-000-003; Jackson ImmunoResearch), anti-rabbit IgG-BV421 (clone Poly4064; BioLegend), and anti-human IgG Fcγ-Alexa Fluor 647 (709-606-098; Jackson ImmunoResearch). Biotinylated Abs were detected using streptavidin-APC-Cy7 (405208; BioLegend).

## IFM

The IFM was performed as previously reported (Nihei et al, 2023a). Briefly, kidneys excised from mice were directly embedded in OCT compound (Sakura Finetek), frozen in liquid nitrogen, and stored at −80°C. Frozen specimens were sliced to make 6-μm sections and fixed with ice cold acetone at −30°C for 1 min. After blocking, the sections were incubated with primary Abs for 60 min, followed by incubation with secondary Abs for 60 min. The samples were examined using a fluorescence microscope (BZ-9000; KYENCE) or a laser confocal microscope (Nikon; A1 HD25). Where indicated, the mice were transcardially perfused with PBS before kidney excision (Figs 2C and E, 3A, and 4A). For quantification of Ab deposition, the maximum intensities of Ab deposition in each glomerulus were quantitated using ImageJ.

## Histological analysis

Kidneys excised from transcardially perfused mice were fixed in 4% PFA and embedded in paraffin, from which 3-μm-thick sections were prepared. After staining the sections with periodic acid–Schiff, the number of nuclei per glomerulus was counted. The samples

were examined under a fluorescence microscope (BZ-9000; KYENCE).

## BUN analysis

BUN concentration was analyzed using a Fuji-DRY-CHEM 7000V (Fujifilm).

## WB analysis

WB analysis was performed as previously reported (Haniuda et al, 2016). Briefly, mesangial cells were lysed with RIPA buffer (40 mM Tris–HCl, 150 mM NaCl, 1% NP-40, 1% sodium deoxycholate, 0.1% SDS, and 10 mM NaF) containing protease inhibitors. HEK293T cells and bacteria were lysed with 1× sample buffer (50 mM Tris–HCl, 5 mM EDTA, 2% SDS, and 10% glycerol) containing protease inhibitors. Lysates were mixed with the SDS sample buffer, boiled, and used for SDS–PAGE. Proteins on the blotted membrane were detected with serum and/or Abs as follows: For use as primary Abs, mouse sera were diluted at 1:40 and culture supernatant and rmAbs were diluted to the indicated concentration. Anti-mouse IgA Abs conjugated with HRP (1:5,000; Southern Biotech) or anti-human IgG Abs conjugated with HRP (1:8,000; Jackson ImmunoResearch) were used as the secondary Abs. For the detection of FLAG-tagged proteins, rabbit anti-FLAG tag Ab (1:2,000; MBL) and anti-rabbit Ab conjugated with HRP (1:5,000; Jackson ImmunoResearch) were used as the primary and secondary Abs, respectively.

## Immunoprecipitation

Immunoprecipitation was performed as previously described (Nihei et al, 2023a). Briefly, cultured MCs from gddY mice were lysed and incubated with rmAb-immobilized beads (see below). The precipitates were then subjected to SDS–PAGE. rmAb-immobilized beads were generated by coupling rmAb with protein G Sepharose (Cytiva).

## Mass spectrometry analysis

Mass spectrometry analysis was performed as previously reported (Nihei et al, 2023a). Briefly, the gel was stained using a Pierce silver stain for mass spectrometry kit (Thermo Fisher Scientific) and the regions of interest were excised from the silver-stained gel. After destaining, the excised bands were digested using trypsin/Lys-C (Promega). The peptides were identified by nano–LC-MS/MS on a Triple TOF 5600+ system operated with Analyst TF 1.7 software and an Eksigent nano LC system (AB SCIEX, MA). The obtained MS data were searched using ProteinPilot 5.0.1 Software (AB SCIEX) using the UniProt database (2020_06). A confidence cut-off of 1% false discovery rate was used for protein identification.

## Plasmid constructions

cDNAs for mouse *CBX1*, *CBX3*, *CBX5*, and *CBX3* 1/2 or 2/3 were generated by PCR using RNA from cultured MC of gddY mice with KOD Fx Neo or KOD plus neo DNA polymerases (Toyobo), and subcloned into a p3xFLAG-CMV-10 (Sigma-Aldrich) vector to produce proteins with N-terminal FLAG-tags. Human *CBX3* cDNA was

generated as described above using RNA from HEK293T cells, subcloned into p3xFLAG-CMV-10, or fused with linker peptide (G4S) and Strep-tag**II** (WSHPQFEK) at the C terminus, and subcloned into a pcDNA3.4-TOPO vector (Thermo Fisher Scientific). To produce recombinant CBX3 in bacteria, *CBX3* cDNA was subcloned into a pTrcHis2-TOPO vector (Sigma-Aldrich) containing a C-terminal 6× His-tag. To induce alanine mutations in CBX3 1/2, overlapping primers, each carrying a mutated codon, were used with either a primer at the 5' or 3' end of the cDNA fragment for the two-step PCR. The primers used are listed in Table 1.

## ELISA

ELISA was performed as previously described (Nihei et al, 2023a). For detection of urinary albumin, microtiter plates were coated with 50 $\mu$l of anti-mouse albumin Ab (Bethyl Laboratories), diluted to 1 $\mu$g/ml with 0.1 M sodium bicarbonate (WAKO). The plates were serially incubated with 50 $\mu$l of diluted urine samples (1:10,000 for urine from gddY mice and 1:100 for urine from BALB/c mice), 50 $\mu$l of biotinylated anti-mouse albumin Ab (Bethyl Laboratories), and 50 $\mu$l of streptavidin conjugated with HRP (1:3,000; Southern Biotech), each at RT for 1 h. For the detection of anti-CBX3 IgA in human sera, the 384-well plates were coated with 30 $\mu$l of human recombinant CBX3 (10 $\mu$g/ml). The plates were serially incubated with 30 $\mu$l of diluted sera (1:100) at RT for 2 h and 30 $\mu$l of goat anti-human IgA1/A2 Abs conjugated with HRP (1:8,000; Abcam) at RT for 1 h. Other Abs used were as follows: rmAbs from gddY mouse kidney PBs (1 $\mu$g/ml for plate coating), anti-FLAG rabbit Ab (1:2,000; MBL), and anti-rabbit Ab conjugated with HRP (1:5,000; Jackson ImmunoResearch). The plate-bound HRP reacting with 3,3',5,5'-tetramethylbenzidine was evaluated by measuring the absorbance at 450 nm. Recombinant mouse CBX1, CBX3, and CBX5 proteins coated on plates at 50 $\mu$g/ml were blocked with RPMI 1640 medium containing 3% FCS, detected with the rmAbs (0.66 $\mu$g/ml) and HRP-conjugated anti-human IgG (1:5,000; Jackson ImmunoResearch) and developed with BioFX TMB super sensitive (SURMODICS).

## Preparation and culture of cells and FCM

Cell preparation and culture, FCM, and single B-cell sorting from mice were performed as previously reported (Nihei et al, 2023a). Briefly, leukocytes in the kidney and salivary glands were isolated using Percoll gradient centrifugation. Mononuclear cells in the lamina propria of the small intestine were isolated, as previously described (Luck et al, 2019). These cells were cultured in "B-cell medium" (RPMI 1640 [Wako] supplemented with 10% FBS [Biological Industries], 10 mM HEPES [Invitrogen], 1 mM sodium pyruvate [GIBCO], 5.5 × $10^{-5}$ M 2-ME [Invitrogen], 100 U/ml penicillin, and 100 $\mu$g/ml streptomycin [GIBCO]). The culture supernatants used for WB or FCM were diluted with the same medium to adjust the IgA concentration to 1 $\mu$g/ml. Mouse MCs isolated from the glomeruli (Hochane et al, 2013) were cultured in DMEM (high glucose, with L-glutamine and sodium pyruvate; WAKO) with 10% FBS, 10 mM HEPES, and 100 U/ml penicillin, and 100 $\mu$g/ml streptomycin. The MCs were used between passages 8 and 16. For FCM analysis of MCs, endothelial cells and podocytes, glomerular cells were isolated

from combined kidneys of two to five mice, as described previously (Nihei et al, 2023a).

For the binding analysis of serum IgA and rmAb to bacteria, cultured bacteria (2 × $10^6$ cells) were suspended in PBS containing 1% BSA, mixed with various Ab solutions (containing 1 $\mu$g/ml IgA) or rmAb (1 $\mu$g/ml), and left overnight on ice. After washing with PBS containing 1% BSA, bacteria-bound IgA or rmAb was detected using anti-mouse IgA Ab-APC or anti-human IgG Fc$\gamma$-Alexa Fluor 647, respectively. Bacteria were also stained with SYBR Green I Nucleic Acid Gel Stain (Takara Bio). For the binding inhibition assay, equal volumes of CBX3, OVA (77120; Thermo Fisher Scientific), or CGG (D602-0100; Rockland) proteins at various concentrations were incubated for 20 min at RT with rmAb#66 (1 $\mu$g/ml) before incubation with bacteria. The samples were analyzed using FACS Aria II or FACS Canto II (BD Biosciences). The data were analyzed using FlowJo software (Tree Star).

## Generation of rmAbs

The rmAbs were generated as previously reported (Nihei et al, 2023a). Briefly, cDNA was synthesized using total RNA extracted from single IgA$^+$ PBs sorted from the kidneys of gddY mice. The amplified variable regions of the IgA heavy chain and Ig$\kappa$ or Ig$\lambda$ light chain genes were ligated with the pCAGGS expression vector (Hitoshi et al, 1991) into which cDNA encoding human C$\gamma$1, C$\kappa$, or C$\lambda$ was inserted. An rmAb specific for 4-hydroxy-3-nitrophenyl acetyl (NP) was generated by inserting a V$_H$B1-8hi heavy chain (Shih et al, 2002) and $\lambda$ light chain gene into the same vectors. For injection into mice, variable region genes of rmAb#66 or rmAb#N9 were recombined with mouse C$\alpha$, C$\kappa$, or C$\lambda$, and the resultant vectors were transfected into HEK293T cells expressing the J-chain (Nihei et al, 2023a). As a control for in vivo injection, an IgA rmAb derived from a randomly selected single IgM$^+$ IgD$^+$ naïve B cell from a gddY mouse spleen was generated. The rmAbs with human Fc$\gamma$ and polymerized rIgA were purified with Protein A (Cytiva) and CaptoL (Cytiva), respectively, according to the manufacturer's instructions.

## Bacterial culture

Bacteria collected from the oral cavity using a sterile cotton swab and those from the feces and small intestine were suspended in PBS. After sedimentation of the debris, the supernatant of the suspension was applied to a brain heart infusion ager (BD Bioscience) and cultured at 37°C in anaerobic conditions prepared with AnaeroPack Box Jar and AnearoPack-Anaero (Mitsui Gas Co.)

## Recombinant protein production

His-tagged CBX3 protein used for mouse immunization (Fig 2D and E) was produced in *E. coli* strain BL21 transformed with pTrcHis2-CBX3 and purified using Ni-NTA agarose (QIAGEN). Expression vectors encoding FLAG-tagged mouse CBX1, CBX3, CBX5, divided CBX3, FLAG-tagged human CBX3, and Strep-tag**II**–tagged human CBX3 were transfected into HEK293T cells by PEI "Max" (Mw 40,000; Polysciences) and lysed with 1% NP-40 lysis buffer (40 mM Tris–HCl, 150 mM NaCl, 150 mM EDTA, 1% NP-40, and 10 mM NaF) 3 d later. For

**Table 1.  Primer information**

| For 16S rRNA gene sequence | | |
|---|---|---|
| **PCR step** | **Name** | **Sequence (5′→3′)** |
| 1st PCR | 1st-341f_MIX | ACACTCTTTCCCTACACGACGCTCTTCCGATCT-NNNNN*-CCTACGGGNGGCWGCAG |
| | 1st-805r_MIX | GTGACTGGAGTTCAGACGTGTGCTCTTCCGATCT-NNNNN*-GACTACHVGGGTATCTAATCC |
| 2nd PCR | 2ndF | AATGATACGGCGACCACCGAGATCTACAC-Index2-ACACTCTTTCCCTACACGACGC |
| | 2ndR | CAAGCAGAAGACGGCATACGAGAT-Index1-GTGACTGGAGTTCAGACGTGTG |
| (NNNNN*: random sequence with 0–5 base) | | |
| **For generation of CBX proteins** | | |
| FLAG-FL-CBX1 | CBX1 Fw EcoRI | ATAATGAATTCGGGGAAAAAGCAAAACAAGAAGA |
| | CBX1 Rv BamHI | ATAATGGATCCCTAATTCTTGTCGTCTTTTTTGT |
| FLAG-FL-CBX3 | CBX3 Fw EcoRI | ATAATGAATTCGGCCTCCAATAAAACTACATTGC |
| | CBX3 Rv BamHI | ATAATGGATCCTTATTGTGCTTCATCTTCAGGAC |
| FLAG-FL-CBX5 | CBX5 Fw EcoRI | ATAATGAATTCGGGAAAGAAGACCAAGAGGACAG |
| | CBX5 Rv BamHI | ATAATGGATCCTTAGCTCTTCGCGCTTTCTTTTT |
| His-FL-CBX3 | CBX3 Fw NcoI | ATAATCCATGGCCTCCAATAAAACTACATTGCAA |
| | CBX3 Rv HindIII | ATAATAAGCTTGTTGTGCTTCATCTTCAGGACAA |
| FLAG-1/2 CBX3 | CBX3 Fw EcoRI | ATAATGAATTCGGCCTCCAATAAAACTACATTGC |
| | 1/2 CBX3 Rv BamHI | ATAATGGATCCTTATTTCCTTTTTGTAC |
| FLAG-2/3 CBX3 | 2/3 CBX3 Fw EcoRI | ATAATGAATTCGCCAGAAGAAAATTTAGA |
| | CBX3 Rv BamHI | ATAATGGATCCTTATTGTGCTTCATCTTCAGGAC |
| **For generation of mutated CBX proteins** | | |
| **PCR step** | **Name** | **Sequence (5′→3′)** |
| 11MA 1st PCR-1 | CBX3 Fw EcoRI | ATAATGAATTCGGCCTCCAATAAAACTACATTGC |
| | 11MA Rv | CTTTCCCGCTTTTTGCAATGTAGTTTT |
| 11MA 1st PCR-2 | 11MA Fw | CAAAAAGCGGGAAAGAAACAAAATGGA |
| | CBX3 Rv BamHI | ATAATGGATCCTTATTGTGCTTCATCTTCAGGAC |
| 11MA 2nd PCR | CBX3 Fw EcoRI | ATAATGAATTCGGCCTCCAATAAAACTACATTGC |
| | CBX3 Rv BamHI | ATAATGGATCCTTATTGTGCTTCATCTTCAGGAC |
| 32VA 1st PCR-1 | CBX3 Fw EcoRI | ATAATGAATTCGGCCTCCAATAAAACTACATTGC |
| | 32VA Rv | TTTTTCTGCCACAAATTCTTCAGGCTC |
| 32VA 1st PCR-2 | 32VA Fw | TTTGTGGCAGAAAAAGTACTGGACCGT |
| | CBX3 Rv BamHI | ATAATGGATCCTTATTGTGCTTCATCTTCAGGAC |
| 32VA 2nd PCR | CBX3 Fw EcoRI | ATAATGAATTCGGCCTCCAATAAAACTACATTGC |
| | CBX3 Rv BamHI | ATAATGGATCCTTATTGTGCTTCATCTTCAGGAC |
| 54FA 1st PCR-1 | CBX3 Fw EcoRI | ATAATGAATTCGGCCTCCAATAAAACTACATTGC |
| | 54FA Rv | ATCTGTGGCCCCCTTCCACTTCAGGAA |
| 54FA 1st PCR-2 | 54FA Fw | AAGGGGGCCACAGATGCTGATAATACT |
| | CBX3 Rv BamHI | ATAATGGATCCTTATTGTGCTTCATCTTCAGGAC |
| 54FA 2nd PCR | CBX3 Fw EcoRI | ATAATGAATTCGGCCTCCAATAAAACTACATTGC |
| | CBX3 Rv BamHI | ATAATGGATCCTTATTGTGCTTCATCTTCAGGAC |

**Table 1.   Continued**

| For 16S rRNA gene sequence | | |
|---|---|---|
| **PCR step** | **Name** | **Sequence (5'→3')** |
| 58DA 1st PCR-1 | CBX3 Fw EcoRI | ATAATGAATTCGGCCTCCAATAAAACTACATTGC |
| | 58DA Rv | AGTATTAGCAGCATCTGTGAACCCCTT |
| 58DA 1st PCR-2 | 58DA Fw | GATGCTGCTAATACTTGGGAACCAGAA |
| | CBX3 Rv BamHI | ATAATGGATCCTTATTGTGCTTCATCTTCAGGAC |
| 58DA 2nd PCR | CBX3 Fw EcoRI | ATAATGAATTCGGCCTCCAATAAAACTACATTGC |
| | CBX3 Rv BamHI | ATAATGGATCCTTATTGTGCTTCATCTTCAGGAC |

Listed are the primers used for 16S rRNA gene sequencing and for generating FLAG-tagged CBX1, CBX3, CBX5; His-tagged CBX3; and FLAG-tagged CBX3 mutants (11MA, 32VA, 54FA, and 58DA).

ELISA (Figs 1D, 7E, and S1B) and competition assays (Figs 5G and 6B and S5), FLAG-tagged proteins were purified with anti-FLAG M2 Affinity Gel (Sigma-Aldrich), eluted with 0.1 M glycine HCl (pH 3.0), and neutralized immediately with 2 M Trizma base (Sigma-Aldrich). For ELISA of human sera (Figs 1F and S1C), Strep-tagII–tagged human CBX3 was purified using Strep-Tactin Sepharose (IBA) according to the manufacturer's instructions. Purification was confirmed by Coomassie brilliant blue staining.

### Immunization of mice

6-wk-old BALB/c mice were immunized subcutaneously (s.c.) with His-tagged CBX3 protein (200 $\mu$g) emulsified with CFA (500 $\mu$g) or CFA alone, and once again 3 wk later. 4-wk-old BALB/c mice were immunized s.c. with UV-killed C42 or *E. faecalis* ($1.5 \times 10^{10}$) and zymosan (4 mg/ml; Zymosan A; Wako) as an adjuvant (Nihei et al, 2023b). 4 wk after the final immunization, sera were collected from mice and used for WB analysis.

### Administration of rmAb

AID-knockout mice were intravenously injected with polymerized rmAbs p-rmAb#66 or p-rmAb#N9 (300 $\mu$g/mouse). Two hours later, the mice were transcardially perfused with PBS, and excised kidneys were directly embedded in the OCT compound, frozen, and stored at –80°C.

### 16S rRNA gene sequencing

Genomic DNA of bacteria in the oral cavity was extracted using a High Pure PCR Template Preparation Kit (11796828001; Roche). The targeted V3-V4 hypervariable region of the bacterial 16S rRNA genes was amplified by two-step PCR using ExTaq HS (Takara Bio) as described previously (Honda et al, 2023). Amplicons were purified using AMPure XP (BECKMAN COULTER) and sequences of 2 × 300 bp were acquired using a MiSeq System (Illumina) with MiSeq Reagent Kit v3 (Illumina). The primers are listed in Table 1. Raw sequence data were processed as described previously (Honda et al, 2023). Briefly, data were trimmed and filtered using the Fastx toolkit (version 0.0.14) and Sickle (version 1.33). Trimmed reads were merged using FLASH (version 1.2.11), and Quantitative Insights into Microbial Ecology 2 (QIIME2) version 2020.8 was used for taxonomic analysis. Chimeric sequences were removed using the dada2 plugin before sorting into operational taxonomic units under the reference Greengene (ver. 13_8) with a 97% identity.

### Statistical Analyses

All statistically analyzed data were derived from biological replicates. Statistical analyses were performed using the GraphPad Prism software (version 6.0; GraphPad Software). Comparisons between two groups were analyzed using *t* test, and those in Fig 1F were analyzed using the Mann–Whitney *U* test. Differences were considered statistically significant at $P < 0.05$.

### Online supplemental material

Fig S1 shows the evaluation of antigen reactivity of rmAb#66 and sera from patients with IgAN. Fig S2 shows the verification of CBX3 expression on mesangial cell surfaces. Fig S3 shows the effects of AAD on albuminuria and of ABX on glomerular IgA deposition in gddY mice. Fig S4 shows the validation of the WB and FCM analyses of bacteria isolated from gddY mice. Fig S5 shows the antigenic commonalities between C42 and CBX3.

## Data Availability

All data are available in the published article and its online supplemental material.

## Supplementary Information

## Acknowledgements

We thank Tasuku Honjo for AID-deficient mice, Haruhiko Koseki for *CBX3^{fl/fl} CreER^{T2}* mice, members of the Research Institute for Biomedical Sciences, Tokyo University of Science, and members of the Department of Nephrology, Juntendo University Faculty of Medicine, for technical advice and comments,

and Peter D. Burrows for critical reading of the manuscript. This research was supported by JSPS KAKENHI Grant Numbers 20H00510 and 20K21533 (D Kitamura) and by AMED under Grant Number JP22ek0109610 (Y Nihei, Y Suzuki, and D Kitamura).

## Author Contributions

M Higashiyama: conceptualization, formal analysis, validation, investigation, visualization, methodology, and writing—original draft, review, and editing.

K Haniuda: conceptualization and methodology.

Y Nihei: conceptualization, investigation, and methodology.

S Kazuno: investigation.

M Kikkawa: investigation.

Y Miura: investigation.

Y Suzuki: resources and supervision.

D Kitamura: conceptualization, supervision, funding acquisition, visualization, and writing—original draft, review, and editing.

## Conflict of Interest Statement

The authors declare that they have no conflict of interest.

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
