## [Reviewer comments · Life Science Alliance]

Life Science Alliance

Oral bacteria induce IgA autoantibodies against a mesangial protein in IgA nephropathy model mice

Daisuke Kitamura, Mizuki Higashiyama, Kei Haniuda, Yoshihito Nihei, Saiko Kazuno, Mika Kikkawa, Yoshiki Miura, and Yusuke Suzuki

DOI: <https://doi.org/10.26508/lsa.202402588>

Corresponding author(s): *Daisuke Kitamura, Tokyo University of Science*

Review Timeline:	Submission Date:	2024-01-10
	Editorial Decision:	2024-01-12
	Revision Received:	2024-01-25
	Accepted:	2024-01-25

Scientific Editor: *Eric Sawey, PhD*

Transaction Report:

Please note that the manuscript was previously reviewed at another journal and the reports were taken into account in the decision-making process at *Life Science Alliance*. Since the original reviews are not subject to Life Science Alliance's transparent review process policy, the reports and author response cannot be published.

January 12, 2024

RE: Life Science Alliance Manuscript #LSA-2024-02588-T

Prof. Daisuke Kitamura
Tokyo University of Science
Research Institute for Biomedical Sciences
Yamazaki 2669
Noda, Chiba 278-0022
Japan

Dear Dr. Kitamura,

Thank you for submitting your revised manuscript entitled "Oral bacteria induce IgA autoantibodies against a mesangial protein in IgA nephropathy model mice". We would be happy to publish your paper in Life Science Alliance pending final revisions necessary to meet our formatting guidelines.

- please be sure that the authorship listing and order is correct
- please upload your main manuscript text as an editable doc file
- please upload your Table in editable .doc or excel format
- please add a callout for Table 1 to your main manuscript text
- please upload your main and supplementary figures as single files
- please add a Summary Blurb/Alternate Abstract and a Category to our system
- please add the Twitter handle of your host institute/organization as well as your own or/and one of the authors in our system
- please add an Author Contributions section to your main manuscript text and the system
- please add a conflict of interest statement to your main manuscript text
- please use the [10 author names et al.] format in your references (i.e., limit the author names to the first 10)
- please add your main, supplementary figure, and table legends to the main manuscript text after the references section

Figure Checks:

-the anti-FLAG loading control blots are the same in Figure 7B and 7C. Please indicate clearly in the legend that the blots are the same because both panels are from the same cell transfection experiment. Also, it seems that the anti-FLAG blot should include 4 panels to match the loading for 1/2, 2/3, 1/2, 2/3.

A. FINAL FILES:

-- Summary blurb (enter in submission system): A short text summarizing in a single sentence the study (max. 200 characters including spaces). This text is used in conjunction with the titles of papers, hence should be informative and complementary to the title. It should describe the context and significance of the findings for a general readership; it should be written in the

present tense and refer to the work in the third person. Author names should not be mentioned.

B. MANUSCRIPT ORGANIZATION AND FORMATTING:

Sincerely,

January 25, 2024

RE: Life Science Alliance Manuscript #LSA-2024-02588-TR

Prof. Daisuke Kitamura
Tokyo University of Science
Research Institute for Biomedical Sciences
Yamazaki 2669
Noda, Chiba 278-0022
Japan

Dear Dr. Kitamura,

Thank you for submitting your Research Article entitled "Oral bacteria induce IgA autoantibodies against a mesangial protein in IgA nephropathy model mice". It is a pleasure to let you know that your manuscript is now accepted for publication in Life Science Alliance. Congratulations on this interesting work.

DISTRIBUTION OF MATERIALS:

Again, congratulations on a very nice paper. I hope you found the review process to be constructive and are pleased with how the manuscript was handled editorially. We look forward to future exciting submissions from your lab.

Sincerely,
